# Teachers' Perceptions and Experiences of Navigating Continuing Professional Development during the COVID-19 Pandemic

Tyrone Dempsey [ID] and Raj Mestry * [ID]

Faculty of Education, University of Johannesburg, Johannesburg 1806, South Africa; tyronedempsey@bcisb.ac.th
* Correspondence: rajm@uj.ac.za

**Abstract:** Using social constructivism as a theoretical framework, we examined the perceptions and experiences encountered by South African teachers of their professional development during the COVID-19 pandemic. Translating teacher experiences into explicit learning contexts for learners remains an ongoing pedagogical challenge. For teachers to be effective, they must be prepared for changes and adapt swiftly to changing conditions. The need for pedagogical mobility to adopt a perspective of introspective teaching practice that engages learners in meaningful learning environments was crucial during the COVID-19 pandemic. Teachers were compelled to change their pedagogical approach; they were obliged to become digitally literate and apply advanced technology that would enable learners to access a wide range of activities and resources. Using a generic qualitative research design within an interpretivist paradigm, data were collected through semi-structured interviews with six teachers and analysed using Tesch's method of coding. We applied a generic qualitative methodology within an interpretivist paradigm to explore teachers' perceptions and experiences. Six teachers, purposefully selected from various schools, participated in this research. Our findings revealed that lack of guidance and support from educational authorities and school management teams negatively infringed on teachers' practice and professional development. However, teachers collaborated with peers and community members, engaged in knowledge-sharing, and adopted a trial-and-error approach to finding solutions. CPD programmes focusing on classroom management and pedagogical skills are highly recommended to foster the development of more proficient teachers.

**Keywords:** continuous professional development; communities of practice; COVID-19 pandemic; digital literacy; pedagogical mobility; teacher agency; South Africa



## 1. Introduction

Globally, COVID-19 rapidly altered the landscapes of various sectors, including the economy [1], healthcare, and education [2]. Teachers and learners were compelled to stay at home during intermittent lockdowns and school closures mandated by governments. The United Nations [3] averred that the pandemic compelled teachers to adapt swiftly to new pedagogies and technologies to ensure that teaching and learning continued, even though education was provided remotely. Online teaching suddenly replaced traditional on-campus or face-to-face classroom delivery [4]. Initially, many teachers and learners were not proficient in using digital platforms, and many parents also lacked the appropriate skills and aptitudes to effectively assist children with digital learning [5]. Undoubtedly, pedagogical mobility became inevitable for teachers, who were compelled to make paradigm shifts [6]. Thus, teachers had to redefine their pedagogy, because online teaching was an entirely new experience for many.

The COVID-19 pandemic also generated significant disparities, particularly regarding race, class, gender, geography, and access to technology [7]. Owing to their remote and marginalised locations, rural and township schools faced difficulties long before COVID-19 emerged, such as inadequate classrooms and limited access to basic services

such as water, electricity, and telephone lines [8]. The pandemic exacerbated existing issues in underprivileged schools, leading to unfeasible teaching conditions. Affluent schools continued their teaching and learning programmes by migrating to online teaching [7]. During COVID-19, many poor schools were unable to continue with teaching and learning due to connectivity problems, acquisition of technological devices (laptops, smartphones, etc.), and the exorbitant cost of procuring data. These poorer schools normally charge lower fees, and some also serve students eligible for free tuition, which leads to these schools struggling to secure similar resources, resulting in stark educational resource disparities [9]. However, where schools could afford to implement online teaching, many teachers faced challenges in using online platforms to prepare and deliver interactive lessons, conduct tests and examinations, or ensure quality assurance in teaching and assessing learners. Despite these barriers, teachers were compelled to devise strategies to use specialised resources (e.g., science laboratories) and to administer practical assessments using online methods. In fact, it was very difficult for some teachers to teach specific subjects (e.g., physical sciences) online, because it required demonstrating practical experiments and hands-on features. Students were able to participate in the demonstrations online, but teachers required additional support to facilitate this process. This additional support included the need for someone to film the demonstrations, ensuring that the necessary materials and resources were readily available, and managing the logistics of conducting experiments remotely. It thus became imperative to provide teachers with professional development during COVID-19.

The barriers to teachers' continuing professional development (CPD), as emphasised in [10], provided a lens to investigate the challenges that teachers experienced and highlighted implications for effective teaching and learning during COVID-19. Added to these challenges, teachers had to deal with the plight of historically disadvantaged learners who needed access to technological devices and internet connectivity [10]. This inevitably resulted in widening the digital divide, where learners in affluent schools could complete their education with ease, while learners in poor schools were left behind [11]. Thus, the need arose for pedagogical mobility, where teachers were compelled to change to online teaching amidst challenges experienced by learners. This necessitated that the teachers engage in professional development. They had to be digitally literate and work with advanced technology, so that they could become au fait with the implementation of innovative and cutting-edge online lessons. Comprehensive CPD programmes for teachers were an essential component to improve teaching and learning [12]. However, the complexity of crafting and implementing teacher CPD during the COVID-19 crisis was ostensible.

The lack of effective professional development for teachers has garnered much attention. Several studies [13,14] have shown that pre-COVID-19 CPD programmes for public school teachers were not effectively provided because the programmes offered by education districts were generic, rather than being custom-made to meet the needs of teachers. There was a justification to determine the needs of teachers and school contexts so that appropriate CPD programmes could be crafted and implemented [15]. Prior to COVID-19, CPD programmes made no provision for abrupt changes in teaching and learning. This pandemic challenged traditional CPD approaches and provided a renewed lens through which teachers' CPD could function during the transition phase.

On reflection, teachers had to be adaptable and innovative to keep abreast with unexpected changes taking place in the educational landscape [16]. Many teachers, particularly those teaching in poorer areas, found it challenging to acquire online professional development opportunities [11] due to poor internet connections, high-cost devices, and data issues. As a result of the transition to remote learning, inequities in professional development have been emphasised, with teachers having to bear the high cost of accessing and engaging with online professional development opportunities [17]. Teachers had to develop strategies to ensure that teaching and learning took place [18], without having to know what the outcomes would be. Thus, there was a need to provide CPD for teachers, but nobody really knew what a supportive course of action should be. Should everyone be provided

with a free device or internet at no cost? Should teachers first be educated to use these devices effectively? Who will finance the cost of procuring devices and data, especially with soaring inflation in the South African economy?

Imants and van der Wal [19] rightly pointed out that changes in teaching practices can only be sustained if teachers actively participate in the change process. More importantly, teachers' CPD is crucial to permit teachers to update their pedagogical knowledge and skills and adapt to new teaching methods. Chen [20] observed that teachers often react adversely to forced changes relating to pedagogical issues such as changing teaching methods. Research by Hargreaves [21] found that teachers often experienced negative feelings when changing their teaching approaches. Furthermore, intermittent school closures have raised concerns about the potential long-term impact on learners' academic performance [22]. It is vital to seek more systematic or just-in-time learning (JITL) [23] for teachers, which would enable them to have access to the appropriate resources and support required to adapt effectively to new educational realities and beyond—the so-called "new normal".

To obtain a better understanding of how CPD featured during the COVID-19 pandemic, it is imperative to determine teachers' perspectives and experiences of CPD during the COVID-19 pandemic. By exploring their views, we can gain insights into how barriers (if any) affected teachers' CPD and the necessity to develop strategies to overcome these barriers.

This research attempted to answer the following main research question: What were teachers' perceptions and experiences of their CPD during the COVID-19 pandemic?

## 2. Literature Review

A literature review was undertaken to identify groundbreaking research that had already been conducted in this field [24]. It also helped identify potential gaps and limitations to improve the quality and credibility of the research addressed in this study [25].

A plethora of research has been undertaken on CPD, but few studies have highlighted teachers' CPD during a crisis (e.g., the COVID-19 pandemic). This pandemic necessitated that we explore research undertaken by reputable scholars in the field [26–29]. This study contributes to the growing body of knowledge on CPD for teachers during crisis situations and highlights teachers' challenges as they rapidly transitioned to online instruction.

### 2.1. Change Management Principles during the COVID-19 Pandemic

Lewin's model of change management [30] was cited in this research to explain how organisations could implement change successfully. Undoubtedly, unplanned changes brought about by COVID-19 affected education. Lewin's three-phase change model [30] helped teachers, as well as SMTs, to understand how to manage change. The first phase, "unfreezing", involves preparing the organisation for change by replacing old ideas and practices. According to Lewin, the first stage is where the organisation's current state is evaluated, and the need for change is identified [31]. The second phase, "movement", is the actual implementation of the change. For example, face-to-face teaching is replaced with online teaching. The third phase, "refreezing", occurs when the changes are solidified and integrated into the organisation's culture and practices to make these the "new normal". COVID-19, however, did not give schools the opportunity to assess the need for change, but the final step, "refreezing", could perhaps assist in solidifying the differences and making the changes a permanent part of school culture. The post-COVID-19 era will probably make use of blended learning rather than pure online learning.

Many teachers are generally reactive—only when something happens will they then adjust their plans, strategies, and paradigms to changing circumstances. The pandemic forced these changes because it generated something other than what is typical or routinely performed [6]. When teachers were confronted with the COVID-19 pandemic, they were then obliged to shift in their mindset and show a willingness to improve their current position. For example, teachers were forced to migrate to online teaching, and many were

at different levels of digital literacy. If they were prepared for any change, this may have enabled them to adapt more readily and effectively. One way that teachers can prepare themselves for change is fully participating in well-structured CPD programmes. Understanding the strengths of digital literacy can assist teachers in breaking down barriers that disrupt their motivation and prepare them for future educational changes. Solomon and Tresman [31] concluded that values, knowledge, beliefs, and behaviour are vital aspects when teachers need to adjust to different obstacles or changes in education. Guskey and Thomas [32] averred that effective change requires teacher-specific professional development programmes to be put in place. Teachers might have the vision, passion, knowledge, and practice, but if they lack experiential learning, change might not occur [33]. Therefore, the changing mindset of teachers is not enough, and teachers need to understand where they require the necessary skills to enhance their practice.

### 2.2. Digital Literacy in Education

The World Economic Forum [34] maintains that many developed and emerging education systems rely on passive learning, primarily explicit instruction and rote learning, rather than activities that encourage critical and individual thinking. Furthermore, the Fourth Industrial Revolution indicates that technology is essential in ensuring learners' success. Digital literacy skills in public schools are not at the required levels determined by the Department of Basic Education. Wade and Mestry [29] and Selwyn [35] agree that using technology in schools could improve the quality of teaching, and that it will be beneficial to have digital literacy skills to improve the quality of teaching. Before the COVID-19 outbreak, the buzzword in education was "blended learning". This concept incorporates face-to-face learning with digital and technological components into teaching [36] Little did we know then that teachers would be compelled to use technology during the pandemic, and that this would become the "new normal" [37].

It has become increasingly important for teachers to stay abreast with the constantly changing definitions and requirements of digital literacy [38]. Digital literacy encompasses the technical ability to use software and digital devices, along with a wide range of cognitive, sociological, and emotional competencies. Workshops, training, and professional development sessions on digital literacy were not readily available for teachers during the COVID-19 pandemic [29]. Furthermore, Mishra and Koehler [39] and Paudel [6] asserted that simply incorporating technology or digital skills into the teaching process is insufficient. Martin and Grudziecki [40] concur that digital literacy encompasses more than just digital abilities and technical proficiency. The focus should not only be on choosing meaningful technology, but also on considering pedagogical issues for the development of learners during specific learning efforts. This requires teachers to give serious consideration to pedagogical mobility. Teachers must become digitally literate to enable them to teach effectively in order to attain high learner performance. They should know how to integrate technology into their classroom instruction successfully [6].

According to Martin and Grudziecki [40], digital literacy enables teachers to become innovative and creative in their pedagogy while stimulating significant changes in the educational domain. We concur with Srinivasacharlu [41] that teachers need CPD to equip themselves with the digital skills and knowledge needed to prepare them (and learners) for constant educational changes in the 21st century. Mahaye [16] agrees that teachers must become innovative to stay current with changes in the academic environment. Practically, teachers' needs in digital literacy are diverse; therefore, a "one-size-fits-all" approach to professional development is discouraged. Researchers concur that very few schools conducted focused professional development programmes on digital literacy during the COVID-19 pandemic [29,37]. Most teacher development programmes were mainly centred on accessing platforms such as Zoom, Moodle, or Blackboard.

## 2.3. Designing and Implementing CPD

Day and Sachs [42] underscored the importance of CPD for teachers. CPD refers to all structured activities that teachers participate in to improve their practice [42], whether formal or informal [43,44], through internal or external agencies. Furthermore, Bell and Gilbert [45] emphasised that CPD is a dynamic process where teachers constantly seek new skills and techniques to strengthen learners' learning. Some principal objectives of CPD are to encourage teachers to learn, change their traditional beliefs, and acquire new skills that can be used in their everyday work [46,47]. Essentially, appropriate professional development programmes that are aligned to teachers' pedagogical practices enhance teacher performance and, subsequently, improve learner performance.

However, the COVID-19 pandemic presented unforeseen changes, and new educational policies made it difficult for educational authorities or SMTs to plan and implement appropriate CPD programmes for teachers. During the pandemic, priorities shifted to making online teaching and learning a necessity rather than an alternative [48]. The focus was not on school improvement but, rather, on ensuring continuity of education for children. It is critical to explore the different CPD models to investigate teachers' approaches during the COVID-19 pandemic.

We advocate the "action research" model, which provides a means to navigate professional development. Action research includes communities of practice but further explores how teachers gain knowledge themselves [49]. In action research, teachers become researchers to "find" information to gain knowledge or teach themselves appropriate skills. During the COVID-19 pandemic, this is precisely what was required from teachers, but no clear outline was provided to understand how teachers acquire professional development or apply research to teach themselves actively. Burbank and Kauchak [50] proposed that collaborative action research should move away from traditional CPD models, which often impose a passive role on teachers. Teachers are encouraged to view CPD as a process rather than a product of someone else's work. This approach aims to emphasise pedagogical mobility—a shift in the balance of power towards teachers by allowing them to identify and implement the appropriate pedagogy.

## 2.4. Teacher Agency during CPD

McChesney and Aldridge [10] maintain that teacher agencies are crucial in determining teachers' assent to professional development. Teachers appreciate being acknowledged as professionals and having autonomy over their professional learning. However, when this is not recognised, they are more likely to reject new ideas and different approaches presented in professional development programmes. During the COVID-19 pandemic, teachers felt frustrated when professional development content was irrelevant to their needs, or when they were told not to participate in activities that they believed would benefit their professional growth.

According to Geldenhuys and Oosthuizen [51], an ideological viewpoint holds that teachers should have control over their own professional development; however, not all individuals share this perspective. Some teachers may not be aware of current trends in education or of shortcomings in their pedagogy, thus making it difficult for them to choose appropriate professional development programmes. Teachers' vision, attitudes, and dedication are at the heart of CPD [48,51]. Opfer and Pedder [52] concur that schools where collaboration is prevalent tend to produce more positive teacher attitudes towards CPD. However, the issue of contextualising personal vs. collaborative professional development has been an issue for debate [33]. Is a group or an individual more important in school education? Individuals with agency are those who make decisions alone, take initiative, act proactively rather than reactively, and purposefully seek and operate to achieve a particular goal in a specific circumstance [19]. Furthermore, Imants and van der Wal [19] aver that teachers should play an active role in their own CPD. Teachers are the focus of CPD programmes, and they should have the agency to decide what types of CPD programmes they want to participate in.

Collaboration is also crucial in a social constructivist framework because it provides individuals with opportunities to engage in meaningful learning experiences. According to social constructivism, learning is an active and social process where individuals construct knowledge through their interactions with others and the environment [53]. Individuals share a common interest or goal and engage in pedagogical matters through regular interaction and knowledge sharing with colleagues [54]. These individuals share their experiences, perspectives, and expertise and are willing to learn from one another. Through this pedagogical mobility process, individuals or groups construct new knowledge, refine their understanding of the subject matter, and initiate different approaches of delivering these to learners.

### 2.5. Aim and Objectives of the Study

The general aim of this study was to determine teachers' perceptions and experiences of their CPD during the COVID-19 pandemic. This pandemic brought significant changes in the teaching and learning environment, which also affected how teachers' professional development was undertaken. Thus, it is important to understand how teachers perceived and experienced CPD during the pandemic to address any challenges and improve the effectiveness of such programmes.

To achieve the aims of the study, the following objectives were set:

- To explore the nature and essence of CPD programmes by understanding their concept, purpose, and goals.
- To investigate how CPD for teachers was undertaken during the COVID-19 pandemic.
- To determine how CPD programmes might be strengthened amidst unexpected crises to ensure that effective teaching and learning take place.

### 2.6. Methodology

A theoretical framework serves as the cornerstone and foundation for a research study, providing the necessary structure and support for all aspects [55]. This study used social constructivism to frame the research study.

Social constructivism is a learning theory that emphasises the role of social interaction and experience in shaping an individual's knowledge, beliefs, and understanding of the world [56]. In social constructivism theory, learners (in this case, teachers as beneficiaries of CPD) are seen as active participants in the construction of knowledge, closely aligned to pedagogical mobility, rather than as passive recipients of information [57]. The learning process involves constructing meaning through interaction, reflecting on experiences, and building on existing knowledge with other people. Social constructivism is often used in education to explain how learners (or teachers) develop their understanding of a subject through collaboration and interaction with peers and the wider community [58]. In this research, social constructivism is used to explain the interaction and collaboration between teachers, peers, school management teams (SMTs), and officials from education districts in the Department of Education. Social constructivist theory can be applied to various educational settings, including formal and informal CPD. It is for this reason that the researchers used social constructivism as a framework to examine pedagogical mobility, social contexts, and the experiences of teachers during the COVID-19 pandemic, as well as how it shaped their perceptions and experiences of CPD. The methods of learning that took place during the COVID-19 pandemic were unfamiliar, and the historical and situational factors were entirely new for most teachers. Hence, the social constructivist idea of learning provides a lens to explore how teachers collaborated and socially constructed knowledge during the pandemic.

#### 2.6.1. Research Approach, Paradigm, and Design

The research approach adopted for this study was qualitative within an interpretivist paradigm. According to Saunders et al. [59], qualitative research is a suitable approach to determine the experiences and perspectives of teachers with regard to their CPD. Denzin

and Lincoln [60] aver that interpretivism is concerned with understanding individuals' subjective interpretations of their experiences. This approach connects strongly with our interest in understanding how teachers managed obstacles during the pandemic. Moreover, generic qualitative research has investigated people's subjective opinions, beliefs, perceptions,, and experiences [61]. The generic qualitative methodology is particularly suited to interpretivist research, since it enables the collection of rich, in-depth data on individuals' viewpoints and experiences [62]; therefore, the general qualitative approach was deemed the most effective for this study.

This study used a combination of techniques, such as interviews and document analysis, to make meaning of the perceptions and experiences of teachers during the pandemic and how it impacted on their CPD journey.

### 2.6.2. Sampling

Purposeful sampling was utilised to select participants based on specific criteria relevant to the research question [63]. Participants with relevant experience and knowledge to address the research gap were selected [64]. With the assistance of senior education district officials, we identified six participants (teachers) from six different schools located in districts within Gauteng Province who had five or more years of teaching experience and who were likely to have knowledge of the impact of COVID-19 on their professional development. According to Darling-Hammond et al. [65], experienced teachers are more likely to have developed effective strategies for addressing the challenges that they face in the classroom.

### 2.6.3. Data Collection Methods

The primary data collection strategy was semi-structured interviews, due to the versatility of exploring certain topics while allowing for new themes and ideas to be generated [64]. The semi-structured interviews enabled the preparation of some questions in advance to align with the research question [66]. The questions that were asked included the following: What is your understanding of the term continuing or continuous professional development? How did you access professional development opportunities during the pandemic? Can you describe your experience with adapting your teaching methods during the COVID-19 pandemic? Based on your experiences, what recommendations would you offer to educators who are looking to embrace continuous professional development, especially in uncertain circumstances like a pandemic? The interviews allowed the researcher to clarify and probe participants' responses to acquire in-depth responses. Thus, there was continuous probing and follow-up on the participants' responses.

A pilot study was conducted to ensure that the questions were in no way leading, ambiguous, or grammatically incorrect. The semi-structured interview was piloted with a teacher who was not part of the sample. During the piloting phase, grammatical errors and vague questions in the interview schedule were identified and refined. One of these questions was "Can you share an instance where you adopted a novel teaching technique or approach? I'd be interested to learn about the context and the steps you took to implement it".

The interviews were conducted using the Zoom platform, and with the permission of the participants, the interviews were audio recorded. The duration of each interview was approximately 45 min.

### 2.6.4. Data Analysis

The recordings of each interview were transcribed using Trint, a mobile app that transcribes video and audio to text. The documents were then read and reread to ensure the correctness of the transcription. The interviews were only coded after the transcription of all interviews was completed [67]. To analyse data collected from the interviews and document analysis, Tesch's [68] thematic analysis and coding was used to identify patterns and themes in the data [69]. Some of the major themes that emerged included technology

integration (TI), technology readiness (TR), adaptation of teaching methods (PM), support (S), and challenges and barriers to accessing CPD (C). This approach allowed the researchers to identify the key issues and challenges facing teachers in the context of the pandemic, as well as the strategies that they used to navigate the challenges [24].

### 2.6.5. Ethical Considerations

Arifin [70] emphasised the importance of ethical principles in protecting participants in any study. We obtained approval from the university's ethics committee and the Gauteng Department of Education to conduct the empirical research. We also obtained informed consent from the participants and assured them of their anonymity and confidentiality [24]. They were also informed of the purposes of the study and their right to withdraw from the study at any time without prejudice. Permission was obtained from the participants to record the interviews. The transcribed interviews were sent to the participants to verify that the transcriptions were correctly captured. Ethical clearance was also granted by the University of Johannesburg, with ethics number Sem1-2022-067.

### 2.6.6. Trustworthiness

We applied one of Lincoln and Guba's [71] elements of trustworthiness, namely, credibility. To ensure credibility, in-depth interviews were conducted with teachers from diverse school settings [63], and member checking was engaged to verify the accuracy of the researchers' interpretation of the transcribed data. We also engaged in "reflexivity", which involved acknowledging and explaining our personal biases, attitudes, and beliefs that could negatively impact the research [72]. Triangulation was employed to increase the credibility of the results by incorporating data from a review of documents [73]. The use of multiple data sources and methods helped to increase the credibility of the results and provided a more comprehensive understanding of the phenomenon under investigation [73].

### 2.7. Contexts of Study and Biographical Information

The participants' biographical information is shown in Table 1 below.

**Table 1.** Participants' biographical details.

| Participant | Quantile 1–5 | Gender | Subject | Degree | Work Experience (Years) | District D2/D12 |
|---|---|---|---|---|---|---|
| A | 1 | F | Computer applications technology | B.Ed. Senior Phase and FET | 12 | D12 |
| B | 2 | M | Math literacy | BCom and PGCE | 9 | D12 |
| C | 1 | M | Biology | B.Ed. Senior Phase and FET | 6 | D2 |
| X | 5 | F | Economics | Master in Educational Management | 30 | D2 |
| Y | 5 | F | Mathematics | Master in Educational Management | 12 | D2 |
| Z | 5 | F | Biology | Master of Education in Science Education | 7 | D2 |

The South African Council for Educators (SACE) is responsible for professional teachers and oversees the quality, execution, and administration of CPD. Teachers registered with the SACE must earn PD points through approved activities that suit their develop-

mental needs. The policy framework outlines four types of continuous professional teacher development (CPTD) activities: school-driven, employer-driven, qualification-driven, and those offered by approved organisations. It also distinguishes between compulsory and self-selected PD programmes [74].

The study was carried out in Gauteng, South Africa. Gauteng is South Africa's economic centre [75]. The socioeconomic profile of Gauteng is characterised by large juxtapositions, with rich suburbs placed alongside poor townships. Gauteng suffers with several socioeconomic issues, including high levels of unemployment, poverty, and inequity [76]. The location of the research inside Gauteng gives a unique lens through which to investigate these complex socioeconomic processes, shining light on crucial concerns that reverberate both locally and worldwide.

## 3. Results

The data analysis yielded the following themes:

### 3.1. Lack of Clear Guidance and Support for Teachers

Most participants suggested that SMTs and education district officials were unable to provide teachers with the necessary professional support and guidance during the COVID-19 pandemic, largely because of intermittent school closures and regulations on social distancing. Essentially, COVID-19 created pandemonium in the education sector, causing people to take precautions so as not to be infected with the virus. Thus, teachers' professional development was not a priority in many educational institutions. The lack of planning and ad hoc implementation of teachers' professional development during the COVID-19 pandemic emerged as a significant concern.

Participant B was disappointed that CPD had not featured at all in her school:

*They are sorting out our CPD points or whatever. So, but for that first year of COVID, at our school we received nothing from them. I don't think anyone knew what was going on, and I don't blame them. It was during this time we needed guidance and professional help.*

Participant A concurred with Participant B:

*Um, from my HOD, we got no help—nothing. From the education district facilitator, all that was said was that the maximum marks for PAT [Practical Assessment Tasks] for grade twelves will be decreased to 120 marks. These facilitators were not visible and seldom made any contact with teachers at schools in the district. There merely informed us if there were changes made to the assessment regulations. So, no support at all from SMT and education district officials.*

Undoubtedly, this lack of clear guidance and support created significant challenges for teachers during the COVID-19 crisis. Therefore, in this study, the absence of communication and assistance from education authorities resulted in uncertainty, confusion, and anxiety among teachers. Education district officials and SMT members should have taken a proactive stance by setting up helplines to address teachers' concerns regarding curriculum delivery, resources, and digital literacy, especially during the early stages of the pandemic. The transition to online education could have flowed smoothly if teachers were provided with professional development. Proper planning, design, and development of online instructional programmes by the education department would have dispelled apathy, demotivation, and stress if professional development programmes were offered to teachers on a regular basis. However, the sudden shift to online teaching at some schools left school leaders and education district officials totally unprepared to support and guide teachers, because they themselves were not adequately competent to give any kind of support to teachers due to the hasty shift to online teaching impelled by the pandemic.

*3.2. Scarcity and Exorbitant Cost of Resources*

Dube [77], in his study on online learning during the COVID-19 pandemic in rural schools, concluded that many learners in rural areas were excluded due to the unavailability of networks, lack of devices and infrastructure, and insufficient funds to procure data. Moreover, teachers lacked the computer skills, devices, and infrastructure to effectively teach online. These circumstances were more pronounced for learners/teachers in low-income areas, who were already disadvantaged due to the systemic inequality in education [78].

The responses from the participants corroborated this situation and confirmed that no uniformity existed. SMT members and teachers were driven to make their own decisions relating to the types of devices or online platforms that they would use, as well as the pedagogy best suited for online teaching and learning.

Participant Y acknowledged that some schools were better prepared than others:

> *Our school was well-prepared, I would say. We had the advantage of using Moodle for online learning resources. I don't think all schools had this, and in our subject WhatsApp group, everyone did something else. This one used Google, this one Zoom and this one Meet. Some teachers didn't even teach or send any resources during the initial closure. I am not sure how they did this.*

Participant Y worked at an affluent school that had well-structured plans for teaching and learning; however, even within the school, teachers who taught the same subjects seemed to be doing things differently:

> *Um, I gave lessons using PowerPoints and provided the learners with notes and they did it like that. The Mathematics and Accounting teachers preferred using the Zoom platform to teach, but for the juniors like us, we didn't teach online because data and connectivity was a problem.*

The variability in approaches and resources among schools had a significant impact on the provision of quality of education during the COVID-19 pandemic. As Participant Y suggested, some schools were better prepared than others to provide online teaching and learning resources for both teachers and learners. The availability of different learning management systems and online platforms in different schools led to confusion and inconsistency, making it difficult for teachers and learners to adapt to the new teaching and learning methods. It was imperative for SMTs to manage pedagogical mobility in their schools.

Participant A, a teacher in a poor no-fee school, mentioned that the facilitator gave instructions on what teaching and learning should look like without considering the available resources and providing professional development for teachers to deal with these pedagogical issues:

> *She didn't take into account the type of learners that I had nor what their background was. So, it might have worked well in other schools in the district, but it couldn't have worked in mine, unfortunately. My learners didn't have computers at home to do practical activities on Word Document.*

The needs and backgrounds of learners differ significantly, and a one-size-fits-all approach is not a practical solution. Teachers need to consider the needs of their learners and adapt their teaching approaches accordingly, especially during the pandemic, where learners faced serious challenges such as limited access to technology and home learning environments that were not conducive to learning.

Participant A's comment also sheds light on the digital divide in South African education. Different approaches should be followed, and these approaches should depend on the context of the school. Teaching and learning continued for learners in affluent schools. However, the culture of teaching and learning reflected that there was no consistency in the teaching programmes, curricula changes, and learner assessments between education dis-

tricts, schools, or subjects. Inconsistencies may stem from teachers who were not properly equipped to teach the curriculum, or from the ineffective utilisation of resources.

### 3.3. Digital Literacy Challenges Faced by Teachers

Resources for online teaching and learning were not readily available to all learners, especially the poor. Teachers thus faced serious challenges to effectively implement learning programmes. Limited access to technology and internet connectivity, inadequate training and professional development regarding digital literacy provided to teachers, lack of resources and support to effectively implement digital technologies in the classroom, difficulty in integrating digital technologies into the curriculum and teaching practices, and inequitable access to technology and digital resources for learners from disadvantaged communities were issues that teachers had to contend with during COVID-19. It is evident that digital literacy preparedness in the South African context was seriously lacking [29].

Participant Y unequivocally stated the following:

*We didn't receive any proper training to adapt to digital literacy. This was a steep learning curve, and many of us struggled to navigate the various digital tools and platforms that were suddenly thrown upon us. This lack of training caused a lot of confusion for all of us.*

Participant X expressed disappointment and frustration with the lack of training provided by education districts and SMTs when the shift to online teaching replaced face-to-face teaching. Participant C concluded the following:

*It's frustrating that the education district didn't offer teachers comprehensive professional development opportunities on digital literacy during the pandemic. We had to work with the resources we thought was fine. Most of the learners didn't have a computer and those who had, did not know how to use it.*

The challenges faced by teachers during the COVID-19 pandemic resulted in substandard quality of teaching and learning. The sudden closure of schools and the subsequent shift to online learning platforms created difficulties for teachers, particularly for those who were unfamiliar with online teaching. It is essential to address these challenges to ensure that teachers and learners can benefit from the opportunities that digital technology offers.

### 3.4. Collaboration and Support among Teachers and Community Members

It became evident that the only way to turn threats into opportunities was to collaborate with colleagues from the same or other schools and work in collaboration with one another towards a common goal [79]. Despite challenges, teaching and learning had to continue, and teachers had to adapt to the changing nature of their work. The social constructivist learning lens also emphasises collaboration among teachers as an essential aspect of learning. Collaboration allows teachers to co-create knowledge, exchange ideas, and build shared understanding. Teachers who collaborate with one another can bring together their unique perspectives, experiences, and abilities to create a more comprehensive and practical learning experience for themselves and their learners.

Participant X indicated that teachers readily helped one another, sometimes sacrificing their own workloads in favour of colleagues:

*They prepared their lessons and then someone else had to upload it for them. So that was. Yeah, bad for the one that could do it. We have a few colleagues that's very clued up, but they have a lot of work because they have to do their own and they had to have help like four or five other teachers as well. So that, yeah, it was difficult for them.*

Participant X's comments highlight that some teachers were better prepared for online teaching than others, resulting in an uneven workload distribution and responsibilities for those who were digitally literate. Teacher-led solutions that focused on collaboration and support obviated challenges for those who lacked the skills and knowledge. Teachers who were more experienced with online teaching and learning could support their less

experienced colleagues by sharing resources, providing guidance, and offering practical support. Teachers who were not digitally literate took over managerial and administrative responsibilities. In this way, the workload was evenly distributed, and all teachers were supported to provide the best possible education for their learners.

These actions during a crisis confirm the findings of Lin et al. [80], who found that face-to-face (FTF) and online collaborations were equally beneficial in supporting teachers' professional development. The education districts should offer more diverse opportunities to foster collaboration and cooperation among teachers.

### 3.5. Strategies in Finding Practical Solutions

During the COVID-19 pandemic, teachers had no option but to use trial-and-error methods, as described by Bell and Gilbert [47], because going online was a new experience for most of them. In many schools, formal professional development opportunities did not exist, and COVID-19 revealed that the education system in South Africa was unprepared to respond to any crisis. Thus, the pandemic compelled teachers to redefine pedagogical strategies that could be employed in teaching during the COVID-19 era.

Participant Y mentioned the following:

> We also didn't meet for staff meetings, um, because of the fact that we were too many staff members and the area to meet was too small. We had to always think of new ways to communicate and share with each other. The WhatsApp groups from the education district was useful as we could share resources and ideas with one another.

As the participant suggested, teachers had to find new ways to communicate and share ideas and resources, because face-to-face meetings were no longer feasible. The use of technology, such as WhatsApp, was an effective way for teachers to share knowledge, ideas, and resources with one another and to help to bridge the gap created by the pandemic. The COVID-19 pandemic forced teachers to experiment and find new solutions to the online teaching and learning challenges that they experienced. Participant B's suggestion of trial and error in finding solutions highlights the importance of experimentation and knowledge sharing during the pandemic. Teachers faced new and unique challenges that required innovative but realistic solutions, and not all solutions worked in all situations. Teachers had to be willing to experiment, share their experiences and knowledge with their colleagues, and be open to learning from successful endeavours and failures.

Through experimentation, knowledge sharing, and trial and error, teachers found innovative and effective solutions to online teaching and were able to address other pedagogical challenges through collaboration. By working together and sharing their experiences, teachers developed a better understanding of what worked and what did not in different situations, and they refined their approaches to teaching accordingly. Pedagogical mobility was a necessary approach to make the best of difficult teaching and learning contexts.

Teacher agency is another strategy that was accentuated in this research. It is important for teachers to keep abreast with the latest trends in education and, therefore, to take ownership of their own professional development. During the pandemic, it was found that teachers felt that their agency was taken for granted, and that they had to attend compulsory professional development workshops provided by education districts or when SMTs arranged workshops at their schools. Empowering teachers with agency for their own professional development was underscored during the pandemic. It is crucial for teachers to have a say in their own professional development.

Participant A averred that teachers may not always find the training provided by education districts relevant to their needs and interests. This resulted in teacher demotivation and apathy to participate in the CPD programmes offered by education districts. Participant A asserted the following:

> I don't always need the training they want me to attend. Whereas if I had the choice of developing where I want to develop, then I would maybe develop things that I have an interest in.

Participant Y agreed:

*I feel that to professionally develop teachers, the officials must first speak ask them what their needs or concerns are. I'm taking an example, the disability program that I was forced to attend. Yes, it was interesting, but there's a lot of other workshops, like how to apply certain cognitive levels that was a lot more important to me during the Covid than explaining to me about disability issues.*

Teacher agency in professional development is vital, since teachers choose to develop their own skills and knowledge in areas that interest them and that are relevant to their specific teaching contexts. They are not in favour of attending professional development programmes if SMTs and education districts do not consider their needs and school contexts.

## 4. Conclusions and Recommendations

COVID-19 brought unique challenges to the realm of education, prompting educators in South Africa to swiftly adapt their teaching methods and embrace pedagogical mobility for continuous professional development. This study sheds light on the multifaceted experiences of teachers navigating the uncharted waters of remote and online education. The identified themes underscore the need for comprehensive support, equitable resource distribution, digital literacy enhancement, collaborative efforts, and strategic solutions and the integration of JITL [23] to ensure quality education, even in times of crisis.

This study recommends and agrees with previous studies on the need for better preparedness and support for teachers during times of crisis. District officials and school management teams should proactively communicate and provide clear guidance to teachers during times of crisis. The establishment of helplines and regular updates on curriculum delivery, resources, and digital literacy can mitigate uncertainty and foster a sense of direction. As schools continue to grapple with the effects of the pandemic, administrators and policymakers need to prioritise the needs of teachers and provide them with the necessary resources and CPD programmes to ensure that they can effectively support their learners in any learning environment. These CPD programmes should prioritise professional development programmes that focus on enhancing teachers' digital literacy skills, offering comprehensive training and support to help educators effectively integrate digital technologies into their teaching practices. To improve the effectiveness of CPD programmes, it may be necessary to move away from a one-size-fits-all approach and towards more personalised and targeted professional development opportunities. This study recommends that empowering teachers, by involving them in decisions about their own professional development [34], can ensure that teachers attend these CPD programmes. We should tailor training programmes to address their specific needs, interests, and the unique contexts of their schools, and embrace the concept of just-in-time learning (JITL) [23] to provide timely and relevant learning opportunities. This could involve greater collaboration between teachers and professional development providers, along with a greater emphasis on ongoing learning and support rather than accumulating CPD points. Empowering teachers to take ownership of their professional development and providing them with the support and resources they need may improve the overall effectiveness of CPD programmes and better prepare teachers for future challenges. Teachers who are motivated and in control of their learning will have a more significant impact on the education sector.

## 5. Limitations of the Study

The researchers are aware that this study had a limited sample size, consisting of only six participants from schools in Gauteng, South Africa. Thus, while the findings shed light on the experiences of these participants, further research with a larger and more diverse sample is warranted to generalise the conclusions.

**Author Contributions:** Conceptualisation, T.D. and R.M.; Investigation, T.D.; Validation, R.M.; Formal analysis, T.D.; Writing—original draft, R.M. and T.D.; Supervision, R.M. All authors have read and agreed to the published version of the manuscript.

**Funding:** This research received no external funding.

**Institutional Review Board Statement:** Ethical clearance was also granted by the University of Johannesburg, with ethics number Sem1-2022-067.

**Informed Consent Statement:** Informed consent was obtained from all subjects involved in the study.

**Data Availability Statement:** Not applicable.

**Conflicts of Interest:** The authors declare no conflict of interests.

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
