# Peer review of "Teachers’ Perceptions and Experiences of Navigating Continuing Professional Development during the COVID-19 Pandemic"

_education, doi:10.3390/educsci13090933_

Round 1
Reviewer 1 Report
This paper presents a useful research carried out on the CPD opportunities of 6 teachers in the South African context within the COVID-19 pandemic. It is generally well-written and structured. However, there are aspects that need to be improved in order for this paper to be publishable. I will first highlight aspects related to each section of the paper and then provide other general comments in the end.
Introduction
This section is quite good and the authors manage to foreground the research problem and the study. There are, however, some issues that I have come across: (1) please define the word 'poor' when referring to schools (see line 48) - since this paper will be read by an international audience, then please make sure you explain what 'poor' in the South African context means and to what extent such school lacked resources compared to other schools; (2) Lines 107-109 you make reference to CPD that provides teachers with support. I suggest that you refer to research on teachers’ CPD highlighting the need for ‘just-in-time’ support and learning opportunities for teachers targeted towards needs in real-time. See, for example, Calleja et al. (2021). Calleja, J., Foster, C. & Hodgen, J. (2021). Integrating ‘just-in-time’ learning in the design of mathematics professional development. Mathematics Teacher Education and Development, 23(2), 79-101. https://mted.merga.net.au/index.php/mted/issue/view/57
Aims and objectives of the study
This section should come after the literature review section.
Materials and Methods
What does ‘materials’ refer to, and why isn’t this section ‘Methodology’? Also, (1) in this section you would need to include data collection and analysis which appear later on in the paper; (2) section 2.3.1 on 'Research approach, paradigm and design' is not a subsection of the literature
review. This needs to be shifted to the Methodology section.
Literature review
In general, it is a good section. However, in lines 186-191 the authors mention teachers' unpreparedness for change and to be able to adapt. Please note that studies on this have been carried out. In particular, the study by Calleja & Camilleri (2021) shows how teachers navigated their ways during the COVID-19 pandemic when undertaking CPD in the form of lesson study. This study shows that COVID-19 disrupted teachers modus operandi, yet it allowed them to discern new opportunities for learning about digital technology use in lesson study. See Calleja, J. & Camilleri, P. (2021). Teachers’ learning in extraordinary times: Shifting to a digitally facilitated approach to lesson study. International Journal for Lesson and Learning Studies, 10(2), 118-137. https://doi.org/10.1108/IJLLS-09-2020-0058
Context of the study
Needs to be substantiated by adding more information about CPD within the South African context and with specific attention given to the school contexts in which the 6 teachers worked.
Data collection
This section needs to be part of the Methodology section. Also, please include more details about the following: (1) Give a couple of examples of interview questions to show how these align with the research questions (lines 341-342); (2) Provide at least one example of how questions were changed following the piloting (see line 348).
Data analysis
This section needs to be part of the Methodology section. Also, please indicate some codes and the resulting themes - maybe in the form of a table (line 359).
Results
This is a well written and structured section of the paper with good reference to the data. However, lines 412-415 do not fit within the discussion of the theme. They discuss the limitation of the study. As such, I suggest that you move this to the conclusion section.
Conclusion and recommendations
Unfortunately, this section is weak. In my opinion, this section needs to be
re-written to highlight the main contributions to knowledge. Also, the results
needs to be compared and contrasted with literature. The recommendations offered are also not new and do not relate much to the present study.
One would have expected suggestions on how to prepare teachers for the unexpected circumstances and how to help them build an attitude of resilience and an awareness of the importance of self-directed collaborative
learning, so that they are better equipped to take up the challenges of unexpected situations like the COVID-19 outbreak.
General comments
As indicated above, the paper requires some restructuring of sections, particularly (1) the literature review which needs to come after the introduction and (2) the methodology and context of the study.
I would also expect to see more reference to literature as indicated above on the importance of providing teachers with 'just-in-time' learning and support and how disruptions can serve as learning opportunities to teachers if they develop attitudes of resilience and self-directed learning.

Title of paper: I suggest a change - replace 'continuous' with 'continuing'. Also, since in the paper the authors refer to CPD as continuing professional development.
Introduction: (1) In line 74 you state that 'Comprehensive CPD program for teachers was ...' - should this be in the plural?; (2) In line 95 you use 'the correct' - I would avoid such words which seem to imply the availability of one correct solution. Instead, I would say, for example, ‘a supportive’; (3) In line 96, you use 'trained'- I suggest that you replace this by 'educated'.
Reviewer 2 Report
Some of the citations are very dated (over 20 years).

Looks fine.
Reviewer 3 Report
The manuscript wants to deepen an important topic such as the adaptation that teachers had to face during the Covid 19 pandemic.
The manuscript has good potential, however there are some weaknesses and shortcomings.
-The abstract lacks methodological details
-The introduction should also contain a review of the scientific literature
- the difficulties inherent in teaching little emphasize the problems within the teacher-pupil relationship and the emotional dynamics
- the introduction lacks aspects of the context taken into consideration, for example in Italy the ministry of education has provided tablets and PCs free of charge to the entire student population without PCs.
- the objectives of the study are generic, the specific objectives are missing
- it is necessary to describe the context of the study, the schools, etc.
- insert a table with the questions of the semi-structured interview with examples of some answers
- indicate by whom the research group is composed
- there is no information on the protocol number of the approval of the ethics committee
- indicate how and why it was decided to close the selection of participants
- lacks a solid paragraph on policy implications
In general, the whole manuscript must be calibrated within the context of study, the results cannot be generalized.
Fine
Round 2
Reviewer 1 Report
Well done for the changes made based on the suggestions provided.I think you managed to address them successfully.
The only issue I have is your lack of engagement with the literature when discussing the findings in the 'Conclusion and Recommendations' section. While the changes made have improved this section, I think that you need to support your findings by comparing these with literature. What do your findings tell us? How are these similar or different to previous studies? Also, your argumentation for recommending, for example, the integration of JITL opportunities and the integration of digital technologies is not new and this needs to be acknowledged. Hence, in my opinion, this paper could become stronger by showing what your research contributes to the literature - a critical analysis is expected.
Hope this helps.
Reviewer 3 Report
The study presents appreciable improvements, as indicated above, the study needs to be further contextualized, the reader must know where the study was conducted, especially since it is a qualitative and located study. As suggested by Lincon and Guba the contextualization is an essential concept not a minus, so it is not useful to think of generalizing the results with so few participants, it is not a minus in any case. Then insert the context of origin in the title or in the keywords, and insert a paragraph on the socio-economic characteristics of the context where the research was conducted.
-The "limits" paragraph should be placed at the end of the paper.
Some typos are present.
